# Transcriptome Sequencing and Differential Analysis of Ovaries Across Diverse States (Follicular and Non-Follicular Phases)

**DOI:** 10.3390/ani15162436

**Published:** 2025-08-20

**Authors:** Jiabei Sun, Tongliang Wang, Yuheng Xue, Zhehong Shen, Chen Meng, Xinkui Yao, Jun Meng, Jianwen Wang, Hongzhong Chu, Wanlu Ren, Linling Li, Yaqi Zeng

**Affiliations:** 1College of Animal Science, Xinjiang Agricultural University, Urumqi 830052, China; 18690101281@163.com (J.S.); wtl13639911402@163.com (T.W.); xyh9523@126.com (Y.X.); 18858657690@163.com (Z.S.); chenmeng0330@126.com (C.M.); yaoxinkui@xjau.edu.cn (X.Y.); mengjun@xjau.edu.cn (J.M.); wjw1262022@126.com (J.W.); 13364712998@163.com (H.C.); renwanlu@xjau.edu.cn (W.R.); lilinling@xjau.edu.cn (L.L.); 2Xinjiang Key Laboratory of Equine Breeding and Exercise Physiology, Urumqi 830052, China; 3Horse Industry Research Institute, Xinjiang Agricultural University, Urumqi 830052, China

**Keywords:** Kazakh horse, ovary, anovulatory phase, follicular phase, transcriptome sequencing, differentially expressed genes

## Abstract

The Kazakh horse, a breed native to China, demonstrates outstanding adaptability and unique reproductive characteristics. This study explored the differences in gene expression in the ovaries during the non-follicular phase and the follicular phase, a key stage of the reproductive cycle. Through advanced RNA sequencing technology, we identified 979 differentially expressed genes in ovarian tissue, among which 619 were upregulated and 360 were downregulated during the follicular phase. Key genes such as BMP15, LHCGR, COL1A1, and NTRK2 regulate follicular development through key biological pathways, including ovarian steroid synthesis, PI3K-Akt signaling, and extracellular matrix–receptor interaction. Functional analysis revealed that these pathways affect hormone production, cell proliferation, and follicular structural support. Our research has, for the first time, provided a comprehensive transcriptome map of ovarian activity in Kazakh horses, offering important insights into the molecular mechanisms that control their fertility. This research lays the foundation for improving the reproductive strategies and reproductive management of equine animals.

## 1. Introduction

China is home to 29 indigenous and 13 developed equine breeds, among which the Kazakh horse is a primitive local breed valued for both riding and draft purposes. It is particularly noted for its exceptional endurance, strong adaptability, and distinctive physiological characteristics. Kazakh horses are primarily distributed in the Ili Kazakh Autonomous Prefecture and surrounding regions. They exhibit clear seasonal estrus behavior. As a typical species in Xinjiang, Kazakh horses have developed unique biological traits through long-term natural selection and breeding, including resistance to coarse feed, strong endurance, and high resilience [1].

The ovary is the core reproductive organ of female animals, and its function is closely related to an individual’s growth, development, and reproductive capacity. As the most fundamental structural unit of the ovary, the follicle’s development process includes stages such as primordial follicle, primary follicle, secondary follicle, antral follicle, and preovulatory follicle; it eventually matures and is released [2]. During the embryonic period, oogonia undergo meiosis to form primary oocytes, which are then encapsulated by a single layer of flat granulosa progenitor cells to form primordial follicles, accumulating the ovarian reserve pool. After birth, some primordial follicles are activated. Granulosa cells change from flat to cubic and rapidly proliferate into multiple layers. At the same time, stromal cells differentiate into the inner and outer membranes around the follicles, forming primary and secondary follicles. Subsequently, fluid accumulation occurs in the follicular cavity, developing into antral follicles (tertiary follicles). At this point, granulosa cells further differentiate into two layers: parietal granulosa cells, which constitute the main body of the follicular wall, and corona radiata cells, which closely adhere to the oocyte. The latter, together with the oocyte, form the cumulus complex, and all cell layers maintain close contact through gap junctions [3,4,5]. When the diameter of the follicle increases to approximately 1 mm, its continued growth, differentiation, and steroid hormone synthesis functions are dominated by gonadotropins (FSH and LH) secreted by the pituitary gland. FSH and LH respectively act on the receptors of granulosa cells and membrane cells, driving the dilation of the sinus cavity and the proliferation and differentiation of granulosa cells and promoting the synthesis of androgens by membrane cells and the conversion of them into estrogen by granulosa cells. In addition to FSH/LH, insulin, various growth factors, and cytokines also regulate this process. Ultimately, under the stimulation of the LH peak, a very small number of dominant follicles complete maturation and ovulate, while the vast majority of follicles degenerate and disappear through programmed atresia at various stages of development [6,7]. In recent years, researchers have delved deeply into the molecular network of follicular development and ovulation and found that it is coordinated by highly complex interactions of multiple signaling pathways. Specific genes and hormones can significantly promote follicular development. Zhao Z’s [8] research found that LHCGR, STAR, NTRK1, 3β-HSD, TMEM200A, and CDO1 not only participate in the regulation of follicular maturation but also can significantly increase the birth rate of multiple goats by acting on small follicles. Based on the correlation between the development of follicles in domestic animals and their reproductive performance, in-depth research on this process holds significant theoretical and practical value for clarifying their reproductive characteristics and enhancing production efficiency.

In recent years, transcriptomic technologies have been increasingly applied to studies of animal reproduction, offering new insights into the underlying molecular mechanisms. Transcriptome sequencing enables a comprehensive analysis of gene expression profiles in the ovaries of Kazakh horses in different physiological states, facilitating the identification of differentially expressed genes (DEGs) related to follicular development, corpus luteum formation, and hormone biosynthesis. Changes in gene expression not only reflect the ovarian physiological state but may also influence key regulatory factors affecting reproductive efficiency. However, transcriptomic studies focusing specifically on equine ovarian tissue remain limited. Therefore, investigating gene expression differences in the ovarian tissues of Kazakh horses at different physiological stages and analyzing the associated molecular mechanisms provides an essential foundation for advancing research on equine reproductive performance and developmental biology.

## 2. Materials and Methods

### 2.1. Experimental Animals and Sample Collection

Twelve healthy Kazakh horses, aged 5–8 years, were selected from a slaughterhouse in the Tacheng region of Xinjiang, China. The animals were divided into two groups based on their reproductive status: the anovulatory phase (Group B) and the follicular phase (Group Y), with six horses in each group. Ovarian tissues were collected and frozen in liquid nitrogen and stored in liquid nitrogen for subsequent analysis.

### 2.2. Experimental Methods

#### 2.2.1. RNA Extraction, Library Construction, and Sequencing

Total RNA was extracted using a Trizol reagent kit (Invitrogen, Carlsbad, CA, USA) following the manufacturer’s protocol. The quality of the RNA was assessed using an Agilent 2100 Bioanalyzer (Agilent Technologies, Palo Alto, CA, USA) and verified through RNase-free agarose gel electrophoresis. Upon RNA extraction, eukaryotic mRNA was enriched using Oligo (dT) beads [For prokaryotes: After total RNA was extracted, prokaryotic mRNA was enriched by removing rRNA using the Ribo-ZeroTM Magnetic Kit (Epicentre, Madison, WI, USA)]. Subsequently, the enriched mRNA was fragmented into short fragments using fragmentation buffer and reverse transcribed into cDNA by using the NEBNextUltra RNA Library Prep Kit for Illumina (NEB #7530, New England Biolabs, Ipswich, MA, USA). The resulting double-stranded cDNA fragments underwent end repair, had an A base added, and were ligated to Ilumina sequencing adapters. The ligation reaction was purified with the AMPure XP Beads (1.0×) and subjected to polymerase chain reaction (PCR) for amplification. The resulting cDNA library was sequenced using Illumina Novaseq6000 by Gene Denovo Biotechnology Co. (Guangzhou, China).

#### 2.2.2. Sequence Analysis

mRNA was enriched using magnetic mRNA Capture Beads. The enriched mRNA was fragmented at high temperatures and used as a template to synthesize double-stranded complementary DNA (cDNA). End repair and A-tail addition were performed simultaneously. The cDNA was then ligated, purified, and amplified by PCR to construct sequencing libraries. Libraries were sequenced using the Illumina NovaSeq X Plus platform. All procedures were carried out by Guangzhou Gene Denovo Biotechnology Co.

#### 2.2.3. Selection of DEGs

Correlation analysis, co-expression Venn diagrams, and pattern clustering were performed to assess overall expression patterns and their differences. Intergroup difference analysis was conducted using the DESeq2 software (version 3.21). Genes were considered differentially expressed if they met the criteria of |log_2_(fold change)| ≥ 2 and *p* < 0.05 [9].

#### 2.2.4. Gene Function Enrichment Analysis

Gene-specific gene enrichment analysis was performed using the clusterProfiler package in R. The Kazakh horse genome annotation from the Gene Ontology (GO) and Kyoto Encyclopedia of Genes and Genomes (KEGG) database was used as the background reference. GO terms and KEGG pathways with *p* < 0.05 were considered significantly enriched and used to identify biological processes related to ovarian function [10].

#### 2.2.5. Quantitative Real-Time PCR (RT-qPCR) Validation

To verify the reliability of the transcriptome sequencing results, 11 differentially expressed mRNAs were randomly selected for validation by RT-qPCR. GAPDH and β-actin were used as internal reference genes. Primer sequences are listed in the Appendix A

#### 2.2.6. Protein–Protein Interaction

The Protein–Protein Interaction network was constructed using String v10, designating genes as nodes and interactions as lines. The resulting network file was visualized using Cytoscape (v3.9.1) software to depict core and hub gene biological interactions.

#### 2.2.7. Data Analysis

Data visualization was generated using GraphPad Prism 8. All results are presented as the mean ± standard deviation (SD).

## 3. Results and Analysis

### 3.1. Sequencing Data Quality Control and Mapping

Transcriptome sequencing was performed on 12 ovarian tissue samples from Group B and Group Y. The anovulatory group yielded an average of 42,702,683 raw reads, while the follicular group produced an average of 44,365,734 raw reads. After eliminating reads containing 3′-end adapter sequences and those with ambiguous base information, high-quality clean reads were obtained for all 12 libraries. The Q30 score exceeded 95% for all samples from the 12 libraries. The average proportion of clean reads was approximately 99%, indicating high sequencing quality and reliability (Table 1).

### 3.2. Evaluation of Sequencing Results

The sequencing results demonstrated good consistency in gene expression patterns between the samples within Group B and Group Y (Figure 1A,B).

FPKM was used to normalize gene expression levels across samples, with its result indicating the expression levels of various genes. Since different transcript isoforms of the same gene could have distinct biological functions, the FPKM density distribution provided an overview of overall gene expression within each sample, ensuring that global expression distribution remained constant across conditions. Violin plots generated from all 12 ovarian tissue samples revealed a generally consistent distribution of expression levels between Group B and Group Y (Figure 2A). In the PCA graph, the differences between samples are small, while those between groups are large (Figure 2B).

### 3.3. Gene Transcription, Expression, and Clustering Pattern Analysis

A Venn analysis was conducted on ovarian tissue samples from Group B and Group Y. At a gene abundance threshold of 1, a total of 12,752 genes were co-expressed between the two groups. Additionally, 247 genes were uniquely expressed in Group B, while 400 genes were uniquely expressed in Group Y (Figure 3A).

Hierarchical clustering analysis was performed to further examine the overall gene expression patterns between the groups. The clustering results demonstrate clear intergroup distinctions, supporting the reliability of the differential expression pattern among the two groups (Figure 3B).

### 3.4. Analysis of DifferEntially Expressed Genes

Differential gene expression analysis was conducted using the DESeq2 package. Applying a threshold of |log_2_(fold change)| ≥ 2 and *p* < 0.05, a total of 979 DEGs were identified. Among these, 619 genes were upregulated, including BMP15, ATF3, AMIGO2, BRINP2, SULT1E1, and SULT1E1, while 360 genes were downregulated, including COL1A1, NTRK2, LHCGR, COL11A1, and CYP11A1 (Figure 4A,B).

### 3.5. GO Functional Annotation and KEGG Pathway Enrichment Analysis of DEGs

GO enrichment analysis of the 979 differentially expressed mRNAs revealed significant clustering in 59 functional categories. The top 20 enriched GO terms were primarily associated with BP, including cellular process, single-organism process, biological process, regulation of biological process, metabolic process, response to stimulus, multicellular organismal process, developmental process, positive regulation of biological process, signaling, localization, cellular component organization or biogenesis, and negative regulation of biological process. In the MF category, the most significantly enriched term was binding, while, in the CC category, the top terms included cell, cell part, organelle, membrane, membrane part, and organelle part. KEGG pathway enrichment analysis further identified the top 20 significantly enriched signaling pathways involving differentially expressed mRNAs. These included neuroactive ligand–receptor interaction, protein digestion and absorption, extracellular matrix (ECM)–receptor interaction, ovarian steroidogenesis, parathyroid hormone synthesis, secretion and action, cAMP signaling pathway, staphylococcus aureus infection, regulation of lipolysis in adipocytes, phagosome, PI3K-Akt signaling pathway, dilated cardiomyopathy, IL-17 signaling pathway, amoebiasis, cocaine addiction, arachidonic acid metabolism, calcium signaling pathway, pancreatic secretion, cortisol synthesis and secretion, mucin-type O-glycan biosynthesis, and aldosterone synthesis and secretion (Figure 5A,B).

The analysis of the 979 differentially expressed mRNAs revealed significant enrichment in a total of 59 functional categories. A comprehensive list of enriched GO terms and KEGG pathways is provided in the Appendix A

### 3.6. Protein–Protein Interaction Network Analysis

Through bioinformatics prediction, an interaction network of mRNA regulation related to follicular development in Kazakh horses was constructed (Figure 6A,B).

### 3.7. RT-qPCR Validation

To validate the accuracy of the transcriptome sequencing data, 11 DEGs were randomly selected for real-time quantitative PCR (RT-qPCR) analysis. The results, shown in Figure 7, demonstrate that the expression trends of all 11 genes were consistent between RT-qPCR and RNA-seq, confirming the reliability of the sequencing data and the robustness of the gene expression analysis.

## 4. Discussion

In recent years, transcriptomic sequencing technologies have been widely applied in reproductive biology, offering an effective means to explore the molecular mechanisms underlying physiological differences between the anovulatory phase (Group B) and the follicular phase (Group Y). The application of ovarian transcriptomics in equine species enables the identification of DEGs involved in follicular development, thereby contributing to improved reproductive performance and supporting the advancement of equine breeding technologies. Follicles are the fundamental functional units of the female reproductive system, composed of theca cells, granulosa cells (GCs), and oocytes. Their growth and maturation are critical for maintaining the follicular microenvironment, which directly influences female gametogenesis, cyclic hormone secretion, and the regulation of reproductive endocrinology [11]. The process of follicular development is tightly regulated by the interplay of gonadotropins, steroid hormones, and growth factors, which activate multiple intracellular signaling pathways essential for ovarian function. In mammals, various signaling pathways are known to be involved in the regulation of follicular development. In this study, we conducted KEGG pathway enrichment analysis on the identified DEGs (ranked by ascending q-values and *p*-values) and selected the top 20 significantly enriched pathways. These pathways were further examined to reveal the functional interactions and regulatory mechanisms of DEGs involved in ovarian development.

### 4.1. The Process of Follicular Development

Follicular development is a continuous growth process in which its structure undergoes a series of changes. Generally, it can be divided into four stages: primordial follicles, primary follicles, secondary follicles, and mature follicles [12]. Follicular development begins in the embryonic stage. The primordial follicle is composed of an oocyte and a layer of flat granulosa cells (GCs) surrounding it. As the follicle develops, the oocyte enters meiosis and is surrounded by a single layer of granulosa cells, forming a primary follicle. Subsequently, the primary follicle further develops into the secondary follicle, characterized by an increase in the number of granulosa cell layers, the gradual formation of the follicular cavity, and the accumulation of follicular fluid. During this process, the volume of the oocyte increases significantly, and the formation of the follicular cavity marks an important stage in follicular development. In the later stage of follicular development, the follicular cavity is formed, and the oocyte enters the second division of meiosis, preparing for ovulation. During ovulation, the follicle ruptures and the oocyte is released into the fallopian tube to prepare for fertilization. The entire process involves the proliferation, differentiation, and apoptosis of follicular cells, as well as the growth and maturation of oocytes [13,14].

The development of follicles is regulated by various hormones and signaling molecules, such as gonadotropins (FSH and LH), growth differentiation factor (GDF-9), bone morphogenetic protein (BMP), etc. These factors promote the growth and maturation of follicles by regulating the interaction between granulosa cells and oocytes [15,16].

### 4.2. Differential Gene Expression Analysis

In our study, the expression levels of genes related to follicular development, including BMP15, LHCGR, COL1A1, and NTRK2, showed significant differences between Group B and Group Y. Among these genes, BMP15 is a key factor regulating ovarian function [17], which can bind to GCsBMPR-II and ALK6/ALK4 receptors and activate the SMAD transcription factor that regulates gene expression, thereby promoting follicular development [18,19,20,21,22,23,24,25]. QinY [26] et al. found that the fertility of transgenic sows (TG) with the BMP15 gene knocked out significantly decreased. The deletion of BMP15 inhibited the proliferation and differentiation of GCs, leading to abnormal follicular development. At the same time, the development of oocytes was also impaired, resulting in meiosis disorders and a decline in quality. In mice, BMP15 mutations may lead to decreased reproductive capacity or infertility, while, in sheep, homozygotes inhibit follicular development, resulting in infertility [27]. In this experiment, the BMP15 gene was enriched in the ovarian steroidogenesis and cytokine–cytokine receptor interaction pathways. It is speculated that the BMP15 gene binds to ALK4, ALK6, and BMPR-II to activate the TGF-β signaling pathway, thereby affecting follicular development. LHCGR is a core regulatory factor for follicular development and reproductive function [28,29], which can bind to luteinizing hormone and human chorionic gonadotropin to activate the cAMP signaling pathway and thereby promote follicular growth and maturation [30]. Previous studies have demonstrated that IGF-1 enhances the sensitivity of follicles to gonadotropins and promotes estrogen synthesis and follicular selection by up-regulating the expression of FSHR and LHCGR in granulosa cells [31]. Research has found that the expression of LHCGR in mares significantly increases in the later stage of follicular development, especially in pre-ovulatory follicles, indicating the importance of LH in follicular maturation and ovulation [32]. This study found that the LHCGR gene is enriched in the cAMP signaling pathway. It is speculated that after LH/hCG binds to LHCGR, it activates adenylate cyclase (AC) through the Gs protein, increases the cAMP level, and activates PKA. PKA phosphorylates steroids to synthesize the acute regulatory protein (StAR), promoting the transport of cholesterol to mitochondria and initiating the synthesis of progesterone and estrogen [33,34]. COL1A1 is the main constituent subunit of type I collagen in ECM, forming a three-dimensional fibrous network that provides physical support and a mechanical microenvironment for follicles [35,36,37]. Previous studies have found that COL1A1 is an important candidate gene for follicular maturation and ovulation in pigeons [38]. In this experiment, the COL1A1 gene was significantly enriched in the ECM–receptor interaction signaling pathway. It is speculated that the COL1A1 gene activates FAK/PI3K through the ECM–receptor pathway, promoting follicular development. The NTRK2 gene is one of the members of the NTRK family. It mainly binds to BDNF and NT-3 [39], activating downstream pathways to promote follicular growth. Studies have shown that after BDNF binds to NTRK2, the Trk receptor stimulates the PI3K heterodimer, leading to the activation of kinases PDK-1 and AKT. Subsequently, AKT further activates transcription factors such as FRK, BAD, and GSK-3, which affect GC proliferation [40]. After the activation of the PI3K-AKT signaling pathway, the phosphorylation level of AKT protein increases, thereby promoting the proliferation of granulosa cells [41]. In this experiment, the NTRK2 gene was significantly enriched in the PI3K-AKT signaling pathway [42]. It is speculated that the NTRK2 gene and BDNF promote follicular development through the PI3K-AKT signaling pathway. In conclusion, the expression of follicular development-related genes BMP15, LHCGR, COL1A1, and NTRK2 may ultimately affect granulosa cell proliferation and follicular selection, thereby influencing follicular development in Kazakh horses.

### 4.3. KEGG Analysis of DEG

Analysis of the KEGG enrichment pathway indicates that the ovarian steroid synthesis signaling pathway, the extracellular matrix–receptor interaction (ECM) signaling pathway, and the PI3K-Akt signaling pathway play significant roles in follicular development. The ovarian steroid synthesis signaling pathway is related to hormone production, luteal phase formation, and reproductive cycle regulation [43,44,45] and plays an important role in follicular development [46]. Lu Tingting et al. [47] found in their research that in black sheep, steroid synthesis rapid regulatory protein can mediate the entry of cholesterol into the inner mitochondrial membrane, regulate the synthesis of progesterone and estrogen, and affect follicular development. The upregulation of CYP19A1 in healthy follicles promotes the formation of pregnenolone, which is a substrate of progesterone and estradiol and may be the main factor promoting follicular development. The results of this study confirm that the differential gene enrichment pathways in Kazakh horses with different ovarian states include the ovarian steroid synthesis signaling pathway, demonstrating its role in the follicular development of Kazakh horses. The core components of ECM mainly include collagen, elastin, fibronectin, and laminin [48]. They participate in cell signal transduction through downstream genes; regulate the proliferation, differentiation, and survival of granulosa cells; and maintain the integrity of follicular structure [49,50]. Young Park E et al. [51] found that acellular porcine matrix was used as a biomimetic ovarian scaffold to control the growth and development of mouse primal follicles. From this, it can be inferred that ECM plays an important role in the follicular development process of Kazakh horses. The PI3K-Akt signaling pathway provides a material basis for follicular growth by inhibiting the TSC1/2 complex, activating the mammalian target of rapamycin complex 1 (mTORC1) and promoting ribosome biosynthesis and protein translation [52]. The PI3K-Akt pathway plays a promoting role in ovarian follicular development, inhibits follicular atresia, and maintains the survival of dominant follicles [53]. The interaction between the PI3K-Akt pathway and ECM is involved in regulating the cell cycle process and ovulation [54]. Wang J et al. [55] found that the differential genes for follicular growth and development in Celle black sheep were significantly enriched in the PI3K-Akt pathway. Similar studies have also been conducted in cattle. Oocyte growth and follicular development depend on participation in cell proliferation and the regulation of the PI3K-AKT signaling pathway. Stimulating PI3K can promote Akt phosphorylation, leading to the survival of bovine follicles and inducing their growth [56]. The differentially expressed genes BMP15, LHCGR, COL1A1, and NTRK2 screened out in this study were also significantly enriched in the ovarian steroid synthesis signaling pathway, extracellular matrix–receptor interaction (ECM) signaling pathway, and PI3K-Akt signaling pathway. This indicates that the ovarian steroid synthesis signaling pathway, the extracellular matrix–receptor interaction (ECM) signaling pathway, and the PI3K-Akt signaling pathway may affect the follicular development of Kazakh horses by regulating follicular development-related genes.

## 5. Conclusions

This study performed transcriptome sequencing of ovarian tissues from Kazakh horses in different physiological states. The results identified several key DEGs, including BMP15, LHCGR, COL1A1, and NTRK2, which are enriched in ovarian steroidogenesis, PI3K-Akt signaling, and ECM–receptor interaction pathways. These genes are likely to play important regulatory roles in ovarian function.

## Figures and Tables

**Figure 1 animals-15-02436-f001:**
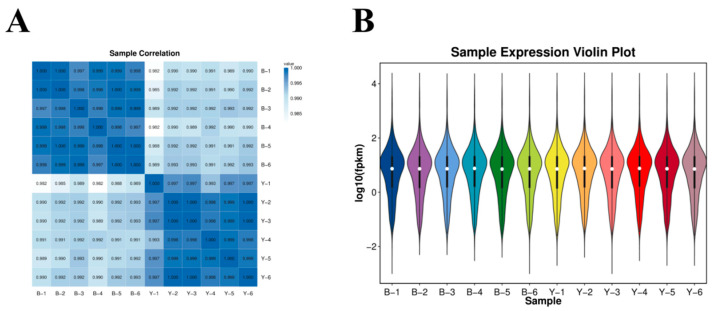
Pearson correlation analysis plot and violin plot of gene expression levels among samples. (**A**) Pearson correlation analysis plot among samples; (**B**) violin plot of gene expression levels. Note: In (**A**), the color intensity of the squares indicates the degree of correlation (R^2^), with darker colors representing stronger correlations closer to 1. In (**B**), the x-axis represents sample names, while the y-axis represents log_10_(FPKM) values. The central line within each box denotes the median gene expression level. Group B corresponds to the anovulatory phase, while Group Y corresponds to the follicular phase.

**Figure 2 animals-15-02436-f002:**
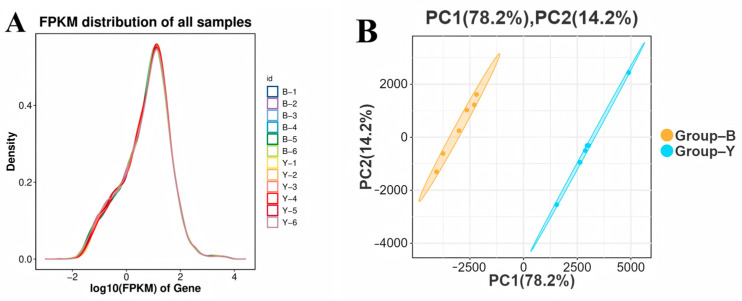
FPKM density distribution plot and principal component analysis plot. (**A**) FPKM density distribution plot; (**B**) principal component analysis plot. Note: In (**A**), the x-axis represents log_10_(FPKM) values and the y-axis shows the density. Each color represents a different sample (B-1 to B-6 and Y-1 to Y-6) and the density curves reflect the distribution of gene expression levels. In (**B**), the x-axis and y-axis represent the first and second principal components, respectively. Groups B and Y exhibited independent, distinct outcomes. Samples within each group indicated low intra-group variability, while samples from Group B and Group Y suggested significant differences in gene expression profiles among groups.

**Figure 3 animals-15-02436-f003:**
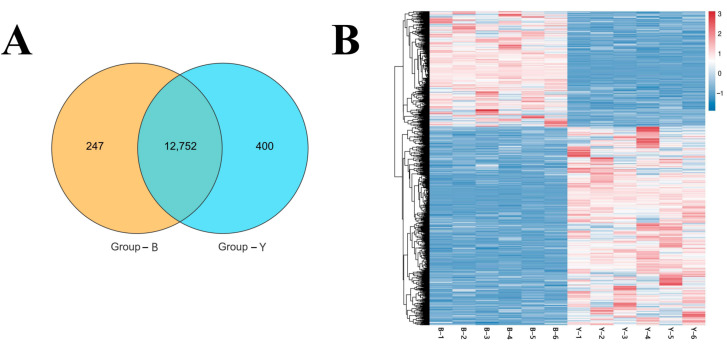
Venn diagram of co-expressed genes and heat map of gene expression patterns between samples. (**A**) Venn diagram of co-expressed genes between two groups; (**B**) heat map of gene expression patterns between samples. Note: In (**B**), each column on the x-axis represents a sample and each row on the y-axis corresponds to a gene. Color intensity reflects relative gene expression levels, with red indicating high expression and blue indicating low expression.

**Figure 4 animals-15-02436-f004:**
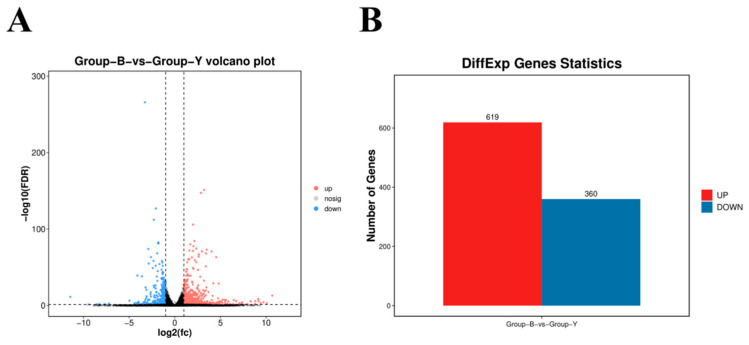
Volcano plot of DEGs between the two groups of samples and bar chart of the number of DEGs. (**A**) Volcano plot of DEGs; (**B**) bar chart of the number of DEGs. Note: In (**A**), the black dotted line indicates the threshold for statistical significance. Red dots represent significantly upregulated genes, while blue dots indicate significantly downregulated genes. In (**B**), the y-axis denotes the number of genes, with the red bar representing upregulated genes and the blue bar representing downregulated genes.

**Figure 5 animals-15-02436-f005:**
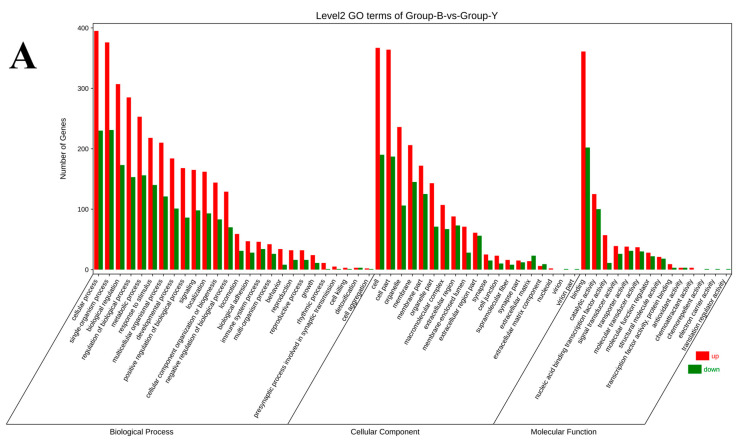
GO classification diagram of DEGs and KEGG pathway enrichment bubble chart. (**A**) GO classification diagram of DEGs; (**B**) KEGG enrichment bubble chart. Note: In (**A**), the x-axis indicates the GO classification, with different sections representing various hierarchies: biological process (BP), cellular component (CC), and molecular function (MF). The y-axis represents the number of enriched genes. In (**B**), the x-axis shows the ratio of DEGs involved in each pathway relative to the total number of DEGs, while the y-axis lists the top 20 significantly enriched pathways. The size of each bubble reflects the number of genes involved in that pathway and the color gradient (red to blue) indicates statistical significance, with red representing more significant enrichment.

**Figure 6 animals-15-02436-f006:**
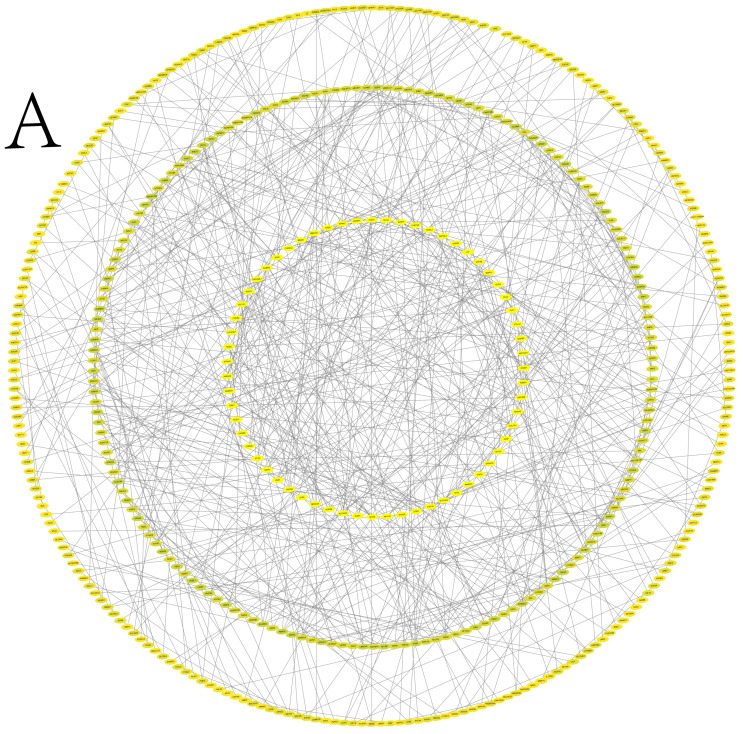
Protein–protein interaction network diagram. (**A**) protein network interaction diagram (for all genes); (**B**) protein network interaction diagram (for core genes).

**Figure 7 animals-15-02436-f007:**
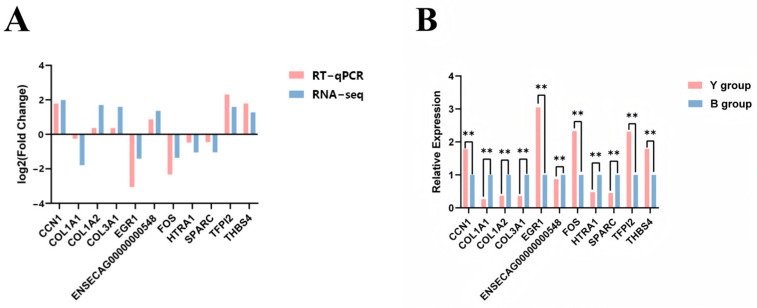
RT-qPCR validation figure. (**A**) The RNA-seq and RT-qPCR Log2FC plot of DEGs; (**B**) relative expression profile of DEGs in RT-qPCR. ** Indicates extremely significant differences.

**Table 1 animals-15-02436-t001:** Transcriptome data quality analysis.

Group	Sample	Raw Reads	Clean Reads	N (%)	Q20 (%)	Q30 (%)	Clean Reads%
B Group	B-1	37,829,074	37,774,698	0.03	98.35	96.00	99.86
	B-2	44,282,326	44,229,932	0.03	98.58	96.51	99.88
	B-3	45,315,170	45,185,390	0.15	98.82	96.83	99.71
	B-4	43,952,288	43,784,598	0.19	98.91	97.15	99.62
	B-5	47,014,742	46,965,604	0.02	98.41	96.15	99.90
	B-6	37,822,500	37,786,314	0.02	98.38	96.10	99.90
Y Group	Y-1	50,051,426	49,991,394	0.04	98.51	96.39	99.88
	Y-2	36,019,928	35,975,982	0.02	98.65	96.68	99.88
	Y-3	43,578,570	43,257,576	0.03	98.50	96.42	99.26
	Y-4	43,430,892	43,270,734	0.20	98.91	97.15	99.63
	Y-5	45,454,976	45,356,876	0.08	98.83	96.90	99.78
	Y-6	47,604,614	47,558,110	0.02	98.28	95.85	99.90

Note: Raw reads refer to the total number of original reads generated. Clean reads denote the number of reads retained after removing low-quality sequences. N indicates the number and proportion of bases identified as “N” in single-end reads relative to the total raw or clean data. Q20 and Q30 represent the percentages of bases with quality scores greater than 20 and 30 (error rates below 1%), respectively.

## Data Availability

The data presented in this study are openly available in BioProject with reference number PRJNA1268976.

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
