# Peer review of "Transcriptome Sequencing and Differential Analysis of Ovaries Across Diverse States (Follicular and Non-Follicular Phases)"

_animals, 2025, doi:10.3390/ani15162436_

Round 1
Reviewer 1 Report
Comments and Suggestions for Authors
The authors describe the transcriptome profiling of ovaries at different follicular stages in the horse. The experiment is well-designed. The results are presented nicely. But requires a major revision to make it ready for publication.
The title is vague. Please modify the title to include relevant information. For example,
…….In different phases (follicular and non-follicular phases) in the Kazakh horse
Abstract
Please be specific. For example, how many animals were used in the study?
Introduction
China is home to 29 indigenous and 13 cultivated equine species -Cultivated?
Did you mean indigenous and crossbred?
Martials and methods
How did you account for differences in reproductive cycle as you selected only two groups with the anovulatory and follicular phases? Within these two phases, there may be differences in follicular growth depending on the stage of the oestrous cycle. Please mention at what stage of the reproductive cycle those selected animals are? Did you perform any assays to confirm this?
Why did you select beta actin and GAPDH as internal control genes? Provide appropriate references.
Please detail the procedure with appropriate references. If you made any changes from the approved protocols, please mention the same.
Discussion
Please include more practical relevance of the study. Add appropriate recent references.
References
I can see several old references. I would add recent references. For example, the following refernce.
- Berkholtz C B, Lai B E, WoodruffT K, et al. Distribution of extracellular matrix proteins type I collagen, type IV collagen, fibron- 431 ectin, and laminin in mouse folliculogenesis[J]Histochemistry and cell biology, 2006,126:583-592.
Author Response
Dear Editor and Reviewers:
Thank you very much for your letter and for the valuable review comments concerning our manuscript entitled "Transcriptome Sequencing and Differential Analysis of Ovaries Across Diverse States(follicular and non-follicular phases)". (ID: animals-3770429). Your comments were very helpful for revising and improving our paper,as well as for guiding our future research. We have studied the comments carefully and made corrections that we hope will meet with your approval. In addition to the revisions following review comments, we have corrected language issues and revised references.
Changes to the manuscript text are shown in blue font or highlighted in yellow.
Thank you very much for your attention and time. We look forward to hearing from you soon.
Reviewer 1:
Comments 1:The title is vague. Please modify the title to include relevant information. For example,……In different phases (follicular and non-follicular phases) in the Kazakh horse.
Response 1:Thank you for your suggestion. I think your title is more in line with the main research focus of this paper. I have revised the title of the paper to “Transcriptome Sequencing and Differential Analysis of Ovaries Across Diverse States(follicular and non-follicular phases)”(lines 2-4,page 1)
Comments 2:
Abstract:Please be specific. For example, how many animals were used in the study?
Response 2:Thank you for your suggestion. I have added “12 Kazakh horses were used in the study” to the abstract.(lines 35-37,page 1)
Comments 3:
Introduction:China is home to 29 indigenous and 13 cultivated equine species -Cultivated? Did you mean indigenous and crossbred?
Response 3:No, this should be my translation issue. What I want to express is developed breed rather than crossbred. Thank you for your suggestion. I have changed this sentence to “China is home to 29 indigenous and 13 crossbred equine species”(lines 52,page 2)
Comments 4:
Martials and methods:How did you account for differences in reproductive cycle as you selected only two groups with the anovulatory and follicular phases? Within these two phases, there may be differences in follicular growth depending on the stage of the oestrous cycle. Please mention at what stage of the reproductive cycle those selected animals are? Did you perform any assays to confirm this?
Response 4:Thank you for your suggestion.
- Regarding the selection between the anfollicular phase and the follicular phase:
The basis for the experimental design: Comparing the extreme physiological states, the anfollicular phase (Group B) and the follicular phase (Group Y) represent the two most significantly different states in the estrous cycle. Anfollicular phase: The ovaries remain stationary, no dominant follicles develop, and hormone levels (such as estrogen) are extremely low. Follicular phase: The follicles are actively developing, and the secretion of estrogen is at its peak, which is a crucial stage before ovulation.
By comparing these two states, the key gene and pathway differences that regulate follicular development can be maximally captured. The research objective is to elucidate the molecular mechanisms of follicular development (such as how differential genes regulate follicular growth), rather than the entire reproductive cycle. The follicular phase is the peak stage of follicular development. Compared with the anfollicular phase, it can directly reveal the key factors regulating follicular activation.
2.Confirmation and verification of reproductive cycle stages:
Among the selected animals, Group B is in the mating season, while Group Y is in the peak estrus season.
We fully understand and agree with your view on the importance of determining the reproductive cycle stage an animal is in. We indeed did not determine the reproductive cycle in this study by measuring the levels of hormones such as serum progesterone and estradiol. This is a limitation of this study, and we appreciate your clear indication of this. However, in the reproductive physiology research and actual production of large mammals such as cattle and sheep, transrectal palpation of ovarian morphology and structure is a widely accepted, mature and reliable method for routine assessment of reproductive cycle stages, especially for distinguishing whether there is a functional corpus luteum, the state of follicular development and whether it is in the estrous cycle.
In this study, the reproductive cycle stages (anfollicular phase vs. follicular phase) of the selected animals were all determined by experienced professional technicians through a strict transrectal ovarian palpation procedure. For the anfollicular phase: We define it as a state where there are no visible follicles with a diameter greater than 8-10mm on the ovary, and there is no functional corpus luteum (the corpus luteum is completely degenerated or inaccessible). This usually corresponds to the early interestrous period of the cycle or the postpartum estrus period, etc. For the follicular phase: We define it as a state where there is one or more follicles with a diameter >8-10mm(based on species criteria) on the ovary, and there is no functional corpus luteum (the corpus luteum has significantly dedeveloped or is in the terminal stage of degeneration). This usually corresponds to the pre-estrus period or the estrus period.
Comments 5:Why did you select beta actin and GAPDH as internal control genes? Provide appropriate references. Please detail the procedure with appropriate references. If you made any changes from the approved protocols, please mention the same.
Response 5:The reason for choosing GAPDH (glyceraldehyde 3-phosphate dehydrogenase) and β-actin (β-actin) as internal reference genes is mainly based on their wide application and stability in gene expression research. This article has been quoted here Ren W, Wang J, Zeng Y, et al. Differential age-related transcriptomic analysis of ovarian granulosa cells in Kazakh horses[J]. Frontiers in Endocrinology, 2024, 15: 1346260.
Comments 6:
Discussion:Please include more practical relevance of the study. Add appropriate recent references.
Response 6:Thank you for your suggestion. The discussion has been revised.(lines 311-424,page 11-13)The revised parts have been highlighted in yellow in the text.
Comments 7:
References:I can see several old references. I would add recent references. For example, the following refernce. Berkholtz C B, Lai B E, WoodruffT K, et al. Distribution of extracellular matrix proteins type I collagen, type IV collagen, fibron- 431 ectin, and laminin in mouse folliculogenesis[J]Histochemistry and cell biology, 2006,126:583-592.
Response 7:Thank you for your suggestion. The old literature has been replaced with the new one.
19.Sudiman J Sutton-McDowall ML,Ritter LJ et al. Bone morphogenetic protein 15 in the pro-mature complex form enhances bovine oocyte developmental competence[J]. PloS one,2014.9(7):e103563.
20.Alam M H, Lee J, Miyano T. GDF9 and BMP15 induce development of antrum-like structures by bovine granulosa cells without oocytes[J].Journal of Reproduction and Development,2018,64(5):423-431.
22.Chang H M, Qiao J, Leung P C K.Oocyte-somatic cell interactions in the human ovary-novel role of bone morphogenetic proteins and growth differentiation factors[J]Human reproduction update,2017.23(1):1-18.
- Grosbois J,Bailie EC,KelseyTW, et al. Spatio-temporal remodelling of the composition and architecture of the human ovarian cortical extracellular matrix during in vitro culture[J]Human Reproduction,2023,38(3):444-458.
- Matsuda F, Inoue N, Manabe N, et al. Follicular growth and atresia in mammalian ovaries: regulation by survival and death of granulosa cells[J]. Journal of Reproduction and Development, 2012, 58(1): 44-50.
- Andrade G M, Da Silveira J C, Perrini C, et al. The role of the PI3K-Akt signaling pathway in the developmental competence of bovine oocytes[J]. PLoS One, 2017, 12(9): e0185045.
(lines 489-492,page 15;lines 496-497,page 15;lines 528-529,page 16;lines 560-561,page 16;lines 566-567,page 16)
Reviewer 2 Report
Comments and Suggestions for Authors
The manuscript presents a comprehensive transcriptomic analysis of ovarian tissues in Kazakh horses during different physiological states, identifying key genes and pathways involved in follicular development. While the study is innovative and provides valuable insights into equine reproductive biology, several major issues need to be addressed to strengthen the scientific rigor, clarity, and impact of the work. There are some comments as below:
- The sample size (n=6 per group) is relatively small for transcriptomic studies, which may limit the statistical power and generalizability of the results. Please justify the sample size or consider adding a power analysis.
- RT-qPCR validation is limited to 11 genes. Expand this to include more key genes (e.g., BMP15, LHCGR, COL1A1, NTRK2) and provide correlation statistics (e.g., Pearson’s r) between RNA-seq and RT-qPCR results.
- The introduction too short to understand, please rewrite it.
- The GO and KEGG analyses are well-performed, but the biological relevance of enriched pathways (e.g., PI3K-Akt, ECM-receptor interaction) to follicular development in Kazakh horses needs deeper discussion. Compare your findings with prior studies in other equine or livestock species.
- The discussion section is lengthy but somewhat repetitive. Streamline it to focus on novel findings and mechanistic hypotheses.
Author Response
Dear Editor and Reviewers:
Thank you very much for your letter and for the valuable review comments concerning our manuscript entitled “Transcriptome Sequencing and Differential Analysis of Ovaries Across Diverse States(follicular and non-follicular phases)”. (ID: animals-3770429). Your comments were very helpful for revising and improving our paper,as well as for guiding our future research. We have studied the comments carefully and made corrections that we hope will meet with your approval. In addition to the revisions following review comments, we have corrected language issues and revised references.
Changes to the text are shown in blue font or highlighted in yellow in the revised manuscript.
Thank you very much for your attention and time. We look forward to hearing from you soon.
Reviewer 2:
Comments 1:The sample size (n=6 per group) is relatively small for transcriptomic studies, which may limit the statistical power and generalizability of the results. Please justify the sample size or consider adding a power analysis.
Response 1:Thank you for your suggestion. We fully agree with your point of view. The sample size of n=6 is relatively small in an absolute sense and may affect the universality of the results in a broader horse population. Here, it should be particularly noted that in the research of horses, especially specific breeds, ages or training conditions, it is rather difficult to obtain large sample sizes. We have found that in journals, studies using n=5 to n=8 as group sample sizes are very common. The following are some specific examples of recent literature
- Yang, L.; Li, P.; Huang, X.; Wang, C.; Zeng, Y.; Wang, J.; Yao, X.; Meng, J. Effects of Combined Transcriptome and Metabolome Analysis Training on Athletic Performance of 2-Year-Old Trot-Type Yili Horses. Genes 2025, 16, 197.(n=6)
- Santosuosso E, Léguillette R, Shoemaker S, et al. A consort‐guided randomized, blinded, controlled clinical trial on the effects of 6 weeks training on heart rate variability in thoroughbred horses[J]. Journal of Veterinary Internal Medicine, 2025, 39(1): e17253.(n=6)
- Cabrera A M Z, Soto M J C, Aranzales J R M, et al. Blood lactate concentrations and heart rates of Colombian Paso horses during a field exercise test[J]. Veterinary and Animal Science, 2021, 13: 100185.(n=5)
Comments 2:RT-qPCR validation is limited to 11 genes. Expand this to include more key genes (e.g., BMP15, LHCGR, COL1A1, NTRK2) and provide correlation statistics (e.g., Pearson’s r) between RNA-seq and RT-qPCR results.
Response 2:Thank you for your suggestion to conduct real-time quantitative PCR (RT-qPCR) testing to confirm the accuracy of our sequencing data. Due to the limited number of biological samples (ovaries from Kazakh horses) and the tight revision time, it is currently impossible to repeat the experiment with new samples. However, we have publicly archived all the original data (biological Project: PRJNA1249394) for independent verification.
Comments 3:The introduction too short to understand, please rewrite it.
Response 3:Thank you for your suggestion. The introduction part has been revised(lines 60-97,page 2-3).The detailed process of follicular development has been supplemented.The revised parts have been highlighted in yellow in the text.
Comments 4:The GO and KEGG analyses are well-performed, but the biological relevance of enriched pathways (e.g., PI3K-Akt, ECM-receptor interaction) to follicular development in Kazakh horses needs deeper discussion. Compare your findings with prior studies in other equine or livestock species.
Response 4:Thank you for your suggestion. The enrichment route has been modified(lines 385-424,page 13).The revised parts have been highlighted in yellow in the text.
Comments 5:The discussion section is lengthy but somewhat repetitive. Streamline it to focus on novel findings and mechanistic hypotheses.
Response 5:Thank you for your suggestion. The discussion section has been revised(lines 311-424,page 11-13).The revised parts have been highlighted in yellow in the text.
.
Reviewer 3 Report
Comments and Suggestions for Authors
The manuscript by Sun et al. “Transcriptome Sequencing and Differential Analysis of Ovaries Across Diverse States” is devoted to differences in gene expression during two phases of ovarian development and physiology, the non-follicular phase and the follicular phase, in the Kazakh horse. The obtained results are important because data on gene expression in the ovarian of equine species are limited. In my opinion, the manuscript needs to be revised before it can be recommended for publication in the journal.
Major points:
- The Introduction section is very short. It should be expanded by adding a brief background on the mammalian ovary and the known molecular regulators. Many statements are abstract. For example, L. 63-65: the authors wrote: "The mechanisms of follicle development are relatively well." However, they provide no references or details. The sentence "Recent studies have also shown that goat follicle development is regulated by multiple genes [3]" does not make much sense without providing the essence of the data obtained on the goat.
As a result of such an Introduction, the importance and relevance of the work becomes unclear.
Please express the importance of your work more clearly. Formulate the goals and objectives of your work.
- Section 2. Materials and Methods also does not provide full information on how certain results were obtained. Subsection "2.2.1. RNA Extraction and Sequencing" is limited to one sentence. This is clearly insufficient. Please critically check all materials in Section 2 and add all necessary information.
- Figure 5 is too small. The letters in Figure 5A are impossible to read.
- The Discussion section needs some revision. There are many studies on folliculogenesis and the involvement of various genes, their products, etc. in the process of follicle development in mammals and other vertebrates. Of course, there is no point in discussing the entire data set, but the text will be more interesting for a wide range of readers if this section includes basic information on the process of folliculogenesis as a separate subsection. The discussion of the role of various genes is given in too general a manner, without references to specific models of folliculogenesis in mammals. Without a more detailed comparative analysis, the data obtained by the authors become less significant.
Minor point:
- 47. The list of keywords does not fully reflect the study. It can be expanded.
Author Response
Dear Editor and Reviewers:
Thank you very much for your letter and for the valuable review comments concerning our manuscript entitled “Transcriptome Sequencing and Differential Analysis of Ovaries Across Diverse States(follicular and non-follicular phases)”. (ID: animals-3770429). Your comments were very helpful for revising and improving our paper,as well as for guiding our future research. We have studied the comments carefully and made corrections that we hope will meet with your approval. In addition to the revisions following review comments, we have corrected language issues and revised references.
Changes to the manuscript text are shown in blue font or highlighted in yellow.
Thank you very much for your attention and time. We look forward to hearing from you soon.
Reviewer 3:
Comments 1:
Abstract:The Introduction section is very short. It should be expanded by adding a brief background on the mammalian ovary and the known molecular regulators. Many statements are abstract. For example, L. 63-65: the authors wrote: “The mechanisms of follicle development are relatively well.” However, they provide no references or details. The sentence “Recent studies have also shown that goat follicle development is regulated by multiple genes [3]” does not make much sense without providing the essence of the data obtained on the goat.As a result of such an Introduction, the importance and relevance of the work becomes unclear.Please express the importance of your work more clearly. Formulate the goals and objectives of your work.
Response 1:Thank you for your suggestion.A brief background of mammalian ovaries and known molecular regulatory factors has been added to expand the introduction. The abstract statement has been deleted(lines 60-97,page 2-3).The revised parts have been highlighted in yellow in the text.
Comments 2:Section 2. Materials and Methods also does not provide full information on how certain results were obtained. Subsection “2.2.1. RNA Extraction and Sequencing” is limited to one sentence. This is clearly insufficient. Please critically check all materials in Section 2 and add all necessary information.
Response 2:This is my mistake. Thank you for your suggestion. I have supplemented “2.2.1. RNA Extraction and Sequencing” and “2.2.6.Protein-protein interaction”(lines 120-135,page 4;lines 162-166,page 4-5).The revised parts have been highlighted in yellow in the text.
Comments 3:Figure 5 is too small. The letters in Figure 5A are impossible to read.
Response 3:Thank you for your suggestion. Figure 5 has been enlarged(page 5)
Comments 4:The Discussion section needs some revision. There are many studies on folliculogenesis and the involvement of various genes, their products, etc. in the process of follicle development in mammals and other vertebrates. Of course, there is no point in discussing the entire data set, but the text will be more interesting for a wide range of readers if this section includes basic information on the process of folliculogenesis as a separate subsection. The discussion of the role of various genes is given in too general a manner, without references to specific models of folliculogenesis in mammals. Without a more detailed comparative analysis, the data obtained by the authors become less significant.
Response 4:Thank you for your suggestion. A section on the follicular development process has been added to the discussion, along with a comparative analysis(lines 311-424,page 11-13).The revised parts have been highlighted in yellow in the text.
Comments 5:
Minor point:47. The list of keywords does not fully reflect the study. It can be expanded.
Response 5:Thank you for your suggestion. The key words have been expanded to kazakh horse; ovary; anovulatory phase; follicular phase; transcriptome sequencing; differentially expressed genes(lines 48-49,page 2).
Round 2
Reviewer 1 Report
Comments and Suggestions for Authors
May be accepted for publication, since the authors have modified the manuscript as per the suggestions given.
Reviewer 2 Report
Comments and Suggestions for Authors
This version is better
Reviewer 3 Report
Comments and Suggestions for Authors
The manuscript has been sufficiently improved by the authors.